# Chamber-Specific Protein Expression during Direct Cardiac Reprogramming

**DOI:** 10.3390/cells10061513

**Published:** 2021-06-16

**Authors:** Zhentao Zhang, Jesse Villalpando, Wenhui Zhang, Young-Jae Nam

**Affiliations:** 1Department of Medicine, Division of Cardiovascular Medicine, Vanderbilt University Medical Center, Nashville, TN 37232, USA; zhentao.zhang@vumc.org (Z.Z.); jesse.villalpando@vumc.org (J.V.); wenhui.zhang@vumc.org (W.Z.); 2Department of Cell and Developmental Biology, Vanderbilt University, Nashville, TN 37232, USA; 3Vanderbilt Center for Stem Cell Biology, Vanderbilt University, Nashville, TN 27232, USA

**Keywords:** reprogramming, fibroblast, cardiomyocyte, chamber

## Abstract

Forced expression of core cardiogenic transcription factors can directly reprogram fibroblasts to induced cardiomyocyte-like cells (iCMs) in vitro and in vivo. This cardiac reprogramming approach provides a proof of concept for induced heart regeneration by converting a fibroblast fate to a cardiomyocyte fate. However, it remains elusive whether chamber-specific cardiomyocytes can be generated by cardiac reprogramming. Therefore, we assessed the ability of the cardiac reprogramming approach for chamber specification in vitro and in vivo. We found that in vivo cardiac reprogramming post-myocardial infarction exclusively induces a ventricular-like phenotype, while a major fraction of iCMs generated in vitro failed to determine their chamber identities. Our results suggest that in vivo cardiac reprogramming may have an inherent advantage of generating chamber-matched new cardiomyocytes as a potential heart regenerative approach.

## 1. Introduction

Heart disease is the leading cause of morbidity and mortality worldwide. A central problem of heart diseases is that cardiomyocytes are irreversibly lost and replaced by cardiac fibrosis. Thus, the cardiac reprogramming approach is particularly attractive in that it directly targets a major source of cardiac fibrosis, cardiac fibroblasts, to induce new cardiomyocytes. It has been shown that the forced expression of core cardiogenic transcription factors (i.e., Gata4, Mef2c, and Tbx5 with or without Hand2) is able to reprogram fibroblasts into induced cardiomyocyte-like cells (iCMs) in vitro and in vivo [1,2,3]. These initial studies provide irrefutable evidence that new cardiomyocytes can be generated by fibroblast to cardiomyocyte fate conversion, proposing an entirely new potential heart regenerative strategy.

There are two different types of working cardiomyocytes in the heart (i.e., atrial and ventricular), each of which are anatomically confined to respective chambers with distinctive functional properties. Atrial and ventricular cardiomyocytes differ significantly in electrophysiological and contractile properties [4]. Ventricular cardiomyocytes have a higher amplitude, longer duration of action potentials and more negative resting membrane potential than atrial cardiomyocytes. These differences in action potentials allow ventricular cardiomyocytes to generate greater contractile force than atrial cardiomyocytes, which have a higher shortening rate than ventricular cardiomyocytes. Thus, ventricles are able to pump out blood against a relatively high-pressure gradient, while atria rather passively send blood to ventricles. Highly orchestrated contraction of these two independent and distinctive types of cardiomyocytes is essential for effective blood circulation. Therefore, a viable heart regenerative strategy should be able to generate chamber-matched working cardiomyocytes in their respective anatomical locations. Otherwise, chamber mis-matched cardiomyocytes may provide a substrate for arrhythmogenesis. For example, after myocardial infarction (MI), which mainly affects the ventricles, ventricular cardiomyocytes need to be regenerated in the affected ventricle(s). However, the ability of the cardiac reprogramming approach to generate a specific type of working cardiomyocytes has not been carefully determined.

In this study, we sought to determine chamber identities of the iCMs generated by either in vitro or in vivo cardiac reprogramming. Using multi-channel high-content imaging, we simultaneously analyzed the induction of ventricular and atrial phenotypes following in vitro cardiac reprogramming. We found that a significant fraction of pan-cardiac marker-expressing cells express both atrial and ventricular markers in vitro. In contrast, in vivo cardiac reprogramming post-MI exclusively induced ventricular-like iCMs in the ventricle. Our results showed successful chamber specification of iCMs by in vivo cardiac reprogramming, in contrast to stochastic and non-exclusive atrial and ventricular specification of iCMs by in vitro cardiac reprogramming. These findings point to an important advantage of in vivo cardiac reprogramming as a potential heart regenerative approach.

## 2. Materials and Methods

### 2.1. In Vitro Cardiac Reprogramming

Retroviruses were generated using the quad-cistronic retroviral vector encoding mouse *Mef2c*, *Gata4*, *Tbx5* and *Hand2* as described previously [5,6]. Briefly, Platinum E cells (Cell Biolabs, San Diego, CA, USA) were transfected with pBabe-X-MGTH construct using Fugene 6 (Promega, Madison, WI, USA). The growth medium (DMEM with 10% FBS and 1% penicillin/streptomycin) on transfected Platinum E cells was changed with the fresh growth medium 16 to 20 h after transfection. The medium containing viruses was filtered through a 0.45 µm polyethersulfone (PES) filter. Polybrene was added to the viral medium at a concentration of 6 µg/mL. MEFs were isolated from wild type mice, and then counted and frozen as described previously [6,7]. After thawing, pre-counted MEFs (5–7 × 10^4^ cells/well) were seeded into a 24-well black clear-bottom plate (Greiner, Monroe, NC, USA, cat# 662892) 16 to 20 h prior to infection. Following removal of fibroblast growth medium, the viral medium was transferred into the cell culture plate containing MEFs. Another transfection was independently performed for generating the second viral medium 24 h after the first transfection. The second viral medium replaced the first viral medium in the culture plate containing MEFs 24 h after the first infection. Twenty-four hours after the second infection, the viral medium was removed from the culture plate containing MEFs, and then replaced with the cardiac induction medium, composed of DMEM/199 (4:1), 10% FBS, 5% horse serum, 1% penicillin/streptomycin, 1% non-essential amino acids, 1% essential amino acids, 1% B-27, 1% insulin-selenium-transferrin, 1% vitamin mixture, 1% sodium pyruvate (Invitrogen), 1 µM SB431542 (Sigma, Darmstadt, Germany), and 0.5 µm A83-01 (Tocris, Bristol, UK) as described previously [6,7]. The induction medium was changed every three days until cells were fixed for immunocytochemistry.

### 2.2. Immunocytochemistry of Reprogrammed Cells In Vitro

Immunocytochemistry was directly performed on a 24-well plate as described previously [5,6,7]. Cells were fixed with 2% paraformaldehyde for 15 min, and then permeabilized with permeabilization buffer (0.05% Triton-X in PBS) for 5 min three times at room temperature. After incubating with blocking buffer (universal blocking buffer, BiogeneX, cat# HK083-50K) for ~45 min at room temperature, cells were incubated with primary antibodies against Troponin I (goat polyclonal, Abcam, Cambridge, MA, USA, cat# 188877, 1:400 dilution), MLC-2v (rabbit polyclonal, Proteintech, Rosemont, IL, USA, cat# 10906-1-AP, 1:200 dilution), and MLC-2a (mouse monoclonal, Synaptic Systems, Goettingen, Germany, cat# 311 011, 1:400 dilution) overnight at 4 °C. Following three 5-min washes with permeabilization buffer, cells were incubated with respective Alexa fluorogenic secondary antibodies (Invitrogen, Waltham, MA, USA) at 1:400 dilution at room temperature for 1 h. Cells were washed again with permeabilization buffer for 5 min three times. During the third wash, DAPI solution at 300 nM (Invitrogen) was added at 1:100 concentration.

### 2.3. High-Content Imaging and Analysis

High-content imaging analysis was performed as described previously [5,6,7]. Briefly, the immunostained cells were analyzed with ImageXpress Micro XL Automated Cell Imaging system (Molecular Devices, San Jose, CA, USA). Cell images were acquired with a 10x objective at 36 fields per each channel per well. DAPI, FITC, Texas Red, Cy5 filter sets were used to detect DAPI, Troponin I, MLC-2v, and MLC-2a, respectively. Thirty-six captured images per each channel per well (a total of 144 images) were analyzed with MetaXpress software (Molecular Devices) to quantify the number of Texas Red, FITC and/or Cy5 fluorescent positive cells among DAPI positive cells. The local background intensity was set to the intensity of a dim cell. The intensity above the local background for each channel was set separately.

### 2.4. Quantitative Real Time PCR (qPCR)

Total RNA was extracted from reprogrammed or uninfected MEFs using a NucleoSpin RNA Kit (Macherey-Nagel, Duren, Germany). cDNA was synthesized by reverse transcription qPCR using a high-capacity cDNA reverse transcription kit (Applied Biosystems, Waltham, MA, USA). qPCR was performed on a Bio-Rad CFX96 system (Bio-Rad, Hercules, CA, USA) using SYBR probes and iTaq Universal SYBR Green Supermix (Bio-Rad).

### 2.5. In Vivo Cardiac Reprogramming

For in vivo reprogramming, retroviruses harboring MGTH or pBabe-X empty vector were generated as described above for in vitro cardiac reprogramming without changing growth media 16 to 20 h post-transfection. The viral medium was collected and filtered through a 0.45 µm polyethersulfone (PES) filter 48 h post-transfection. The viral medium was concentrated with retrovirus precipitation reagent (Alstem, Richmond, CA, USA) as per the manufacturer’s protocol. The concentrated retroviruses were stored at 4 °C and used within two days. Tamoxifen dissolved in corn oil (90%) and ethanol (10%) at a concentration of 50 mg/mL was administered to 10- to 14-week-old Tcf21-MerCreMer (iCre):R26R-tdTomato mice via intraperitoneal injection for five consecutive days. Five days after the last dose of tamoxifen, permanent ligation of the left anterior descending coronary artery (LAD) was performed in Tcf21-iCre: R26R-tdTomato mice to induce MI as described previously [8]. The mice were anesthetized with continuous inhalation of isoflurane without ventilation. A minimal size of left thoracotomy over the left chest was performed. A small skin cut was made over the left chest. After dissecting and retracting pectoral major and minor muscles, a small hole was made at the fourth intercostal space to open the pleural membrane and pericardium. By slightly opening the clamp, the heart was temporarily displaced. The ligation was made at ~3 mm below the origin of the LAD using 6.0 silk. Immediately after LAD ligation, 30 µL of concentrated virus solution was directly injected to a single spot in the border zone of the infarcted left ventricle using a gastight syringe (Hamilton, Reno, NV, USA, cat# 7637-01) and 34-gauge needle (Hamilton, cat# 207434). Intramyocardial infiltration of virus solution was confirmed by observing swelling of the entire anterior wall of the left ventricle. Then, the heart was replaced immediately into the thoracic cavity. After manual evacuation of air, the chest was closed. Buprenorphine (0.05 mg/kg) was subcutaneously injected once before surgery and six times after surgery every 12 h for pain control.

### 2.6. Immunohistochemistry on Heart Sections

After euthanizing the mice by CO_2_ inhalation and cervical dislocation three weeks post-reprogramming, the frozen heart sections were processed. After washing with ice-cold PBS, the dissected hearts were fixed with pre-chilled 4% paraformaldehyde in PBS for 30 min, and then incubated in 30% sucrose in PBS overnight at 4 °C. The fixed hearts were embedded in O.C.T compound and frozen in pre-chilled isopentane. The frozen hearts were sectioned at 8-µm thickness. After 5 min of air drying, the heart sections were washed with PBS three times, and fixed again with pre-chilled 4% paraformaldehyde in PBS on ice for 20 min, followed by PBS washing three times. The fixed heart sections were permeabilized in 0.1% Triton X-100 in PBS for 20 min. Following PBS washing three times, the heart sections were blocked with M.O.M mouse IgG blocking reagent (Vector Labs, Burlingame, CA, USA) for 1 h and 5% goat serum in M.O.M protein diluent (Vector Labs) for 30 min at room temperature. The heart sections were incubated with primary antibodies against Troponin I (goat polyclonal, Abcam, cat# ab56357, 1:400 dilution) and MLC-2v (rabbit polyclonal, proteintech, cat# 10906-1-AP, 1:200 dilution), or MLC-2a (rabbit polyclonal, Proteintech cat# 17283-1-AP, 1:200 dilution) overnight at 4 °C. After washing with PBS three times, the heart sections were incubated with respective Alexa fluorogenic secondary antibodies (Invitrogen) at 1:400 dilution at room temperature for 1 h. tdTomato was visualized without immunostaining. Images were captured using an Olympus IX81 epifluorescent microscope.

### 2.7. Isolation of Cardiomyocytes

Adult mouse cardiomyocytes were isolated from Tcf21-iCre:R26R-tdTomato mice four weeks post-reprogramming using Langendorff perfusion as described previously [9]. The mouse hearts were perfused retrogradely via aortic cannulation in a Langendorff apparatus with three types of buffer solutions: (1) perfusion buffer (NaCl 120.4 mM, KCl 14.7 mM, Na_2_HPO_4_ 0.6 mM, Ka_2_HPO_4_ 0.6 mM, MgSO4 1.2 mM, Na-HEPES 10 mM, NaHCO_3_ 4.6 mM, Taurine 30 mM, BDM 10 mM, Glucose 5.5 mM, pH 7.0), (2) digestion buffer without CaCl_2_ (Collagenase II 2.4 mg/mL in perfusion buffer), and (3) digestion buffer with CaCl_2_ (Collagenase II 2.4 mg/mL, and CaCl_2_ 40 µM in perfusion buffer) [9]. After removing the hearts from the Langendorff apparatus, ventricular cardiomyocytes were mechanically dissociated and triturated using a fine scalpel and scissors and resuspended in stopping buffer (CaCl_2_ 11.7 µM in calf serum 2 mL plus perfusion buffer 18 mL). After centrifuging cells at low speed, the pellet was collected for cardiomyocytes. The isolated cardiomyocytes were fixed with fixation buffer (BD Biooscience, Franklin Lakes, NJ, USA, cat# 554655) for 20 min on ice followed by washing with Perm/Wash buffer (BD Biooscience, cat# 554723) once. The fixed cells were incubated with primary antibodies against MLC-2v (rabbit polyclonal, proteintech, cat# 10906-1-AP, 1:200 dilution) and MLC-2a (mouse monoclonal, Synaptic Systems cat# 311 011, 1:400 dilution) overnight at 4 °C. Following washing with Perm/Wash buffer, cells were incubated with respective Alexa fluorogenic secondary antibodies (Invitrogen) at 1:400 dilution at room temperature for 1 h. After the Perm/Wash buffer wash, the cells were resuspended with stain buffer (BD Biooscience, cat# 554656) containing DAPI solution. The immunostained cardiomyocytes were placed on a glass slide and covered by a glass coverslip. Cardiomyocyte images were captured with a Zeiss LSM 500 confocal microscope.

### 2.8. Statistical Analyses

Statistical significance was determined using one-way ANOVA with Tukey’s post-hoc test or unpaired two-tailed Student’s *t*-test. *p*-values of < 0.05 were regarded as significant.

## 3. Results

### 3.1. Chamber Protein Expression during In Vitro Cardiac Reprogramming

We and others previously observed a heterogeneous population of iCMs, which resemble diverse individual subtypes of cardiomyocytes, following in vitro cardiac reprogramming [10,11]. However, it remains elusive whether atrial and ventricular specifications during cardiac reprogramming are mutually exclusive processes as shown during heart development or whether they are progressed simultaneously. Thus, we sought to determine chamber specification during in vitro cardiac reprogramming using multi-channel high-content imaging analysis. We transduced mouse embryonic fibroblasts (MEFs) with the quad-cistronic retroviral vector encoding four core cardiogenic transcription factors with the splicing order of *Mef2c*, *Gata4*, *Tbx5*, and *Hand2* (referred to as MGTH) as described previously [6]. At three weeks after transduction, we analyzed the induction of a pan-cardiac Troponin I, a ventricular-specific MLC-2v, and an atrial-specific MLC-2a protein (Figure 1). About 40% of cells expressed Troponin I, indicating that these cells adopt a cardiomyocyte fate (Figure 1C, right). While ~26% of the whole population expressed both MLC-2a and Troponin I (~65% of Troponin I^+^ cells), ~16% of cells induced both MLC-2v and Troponin I (~39% of Troponin I^+^ cells) (Figure 1C left). We noted that the induction of an atrial-specific protein is more permissive than ventricular specification during in vitro cardiac reprogramming. Unexpectedly, ~14% of the whole cellular population, which represents most MLC-2v^+^Troponin I^+^ cells, exhibited both atrial and ventricular markers. Only ~2% of the whole population demonstrated ventricular chamber identity without co-developing the atrial phenotype (MLC-2a^−^MLC-2v^+^TnI^+^), while ~12% of the whole population specified atrial chamber identity (MLC-2a^+^MLC-2v^−^TnI^+^) (Figure 1C right). A major fraction of a pan-cardiac marker expressing cells, which are often defined as iCMs, fail to specify their chamber identities during in vitro cardiac reprogramming. These results demonstrated that atrial and ventricular specifications of iCMs can be simultaneously progressed in a non-exclusive manner during in vitro cardiac reprogramming. In addition, we examined the expression of atrial and ventricular genes using qPCR three weeks after MGTH transduction. We found that both atrial (i.e., *MLC-2a*, *Nppa*, *Nppb*, and *Nrtf2*) and ventricular (i.e., *MLC-2v* and *Irx4*) genes are significantly increased by in vitro cardiac reprogramming (Figure 1D). The extent of increase in the expression of atrial genes was relatively greater than the ventricular counterpart.

### 3.2. Chamber Protein Expression during In Vivo Cardiac Reprogramming Post-MI

Next, we examined chamber specification during in vivo cardiac reprogramming post-MI. To identify iCMs derived from cardiac fibroblasts, we used the Tcf21-MerCreMer (iCre):R26R-tdTomato cardiac fibroblast lineage reporter mouse line, which was generated by cross-breeding Tcf21-MerCreMer knock-in mice [2,12] with Rosa26-CAG-LoxP-stop-LoxP-tdTomato mice. In this mouse line, cardiac fibroblasts become labeled with tdTomato upon tamoxifen administration [2]. We directly introduced retroviruses expressing MGTH or control vector into the infarcted left ventricle. At three weeks post-MI, we processed heart sections and immunostained them for pan-cardiac Troponin I and ventricular-specific MLC-2v or atrial-specific MLC-2a (Figure 2A–C). We did not observe any tdTomato^+^Troponin I^+^ iCMs in the control vector-injected hearts. However, we found that MGTH introduction induces a significant number of tdTomato^+^Troponin I^+^ iCMs in the border zone of the infarcted hearts, consistent with the previous studies [2,3,11,13,14] (Figure 2A). Nearly all tdTomato^+^Troponin I^+^ iCMs expressed MLC-2v, while almost no iCMs expressed MLC-2a (Figure 2B). These results demonstrated that ventricular-like iCMs are exclusively generated in the ventricle by in vivo cardiac reprogramming. In addition, we isolated cardiomyocytes using Langendorff perfusion following in vivo cardiac reprogramming. By immunostaining these isolated iCMs for MLC-2v and MLC-2a, we confirmed a ventricular phenotype as well as organized myofibrillar structures of iCMs in a single cell level (Figure 2D). Our findings suggest that the native intra-ventricular environment, in which iCMs are surrounded by continuously contracting native ventricular cardiomyocytes and other types of cells in three dimensions, may provide necessary conditions for ventricular specification which cannot be recapitulated in a plastic dish.

## 4. Discussion

Either atrial or ventricular chamber identity of a cardiomyocyte is acquired in an anatomically defined restrictive location through chamber-specific independent processes during heart development [15]. In contrast, the previous study showed that diverse subtypes of cardiomyocytes (i.e., atrial, ventricular, and pacemaker) are co-induced next to each other by in vitro cardiac reprogramming [10,11]. However, it remains unclear whether atrial or ventricular chamber specification progresses independently or simultaneously in individual iCMs during in vitro cardiac reprogramming. Through this study, we found that both atrial and ventricular chamber specifications can simultaneously progress in individual cells during in vitro cardiac reprogramming. As a result, a significant fraction of iCMs exhibit both atrial and ventricular phenotypes. However, such a hybrid cell does not exist in the post-natal heart. A very small fraction of iCMs showed a pure ventricular-like phenotype without co-expressing an atrial marker, indicating that the induction of a ventricular chamber specification is a less efficient process as opposed to the induction of an atrial phenotype in fibroblasts. This is consistent with our previous study demonstrating the inability of adult fibroblasts to induce a ventricular phenotype, while diverse cardiac phenotypes including ventricular can be induced in MEFs [10]. This uncontrolled chamber specification would be an important obstacle to overcome for the potential in vitro clinical application of directly reprogrammed iCMs in the future. We speculate that successful ventricular chamber specification may require suppression of simultaneously developing atrial specification processes, and vice versa.

In contrast to undetermined chamber specification during in vitro cardiac reprogramming, ventricular-like iCMs were exclusively induced in the ventricle following in vivo cardiac reprogramming post-MI. Ventricular specification of iCMs during in vivo cardiac reprogramming may be analogous to that of native cardiomyocytes in the ventricle during heart development. Our results suggest that an intra-ventricular environment containing native ventricular cardiomyocytes and other ventricular cells may enforce iCMs to adopt a ventricular phenotype. Continuous and vigorous contraction of native ventricular cardiomyocytes as well as secreted molecules from various ventricular cells may be necessary for the ventricular specification of iCMs. In vitro reprogrammed cells may mimic early differentiated cardiomyocytes during heart development, which express both MLC-2a and MLC-2v [16]. Activation of subtype-specific signaling pathways or longer maturation time than in vivo cardiac reprogramming may be needed to induce subtype-specific iCMs in vitro. Despite unsuccessful chamber specification by in vitro cardiac reprogramming, it is important to note that the in vivo cardiac reprogramming approach may have an inherent advantage of generating chamber-matched cardiomyocytes over other heart regenerative approaches using pluripotent stem cell-derived cardiomyocytes, which may have to overcome incomplete chamber specification of newly generated cardiomyocytes. In addition, it would be important to identify ventricular-specific signaling molecules that specify chamber identities of iCMs in an intra-ventricular environment during in vivo cardiac reprogramming for the generation of clinically useful chamber-specific iCMs in vitro.

Our analysis for chamber specification is solely based on the protein expression of chamber-specific proteins. Action potential assessment of iCMs is necessary to accurately define the chamber identity of iCMs. Therefore, the main goal of this study is limited to evaluating chamber-specific protein expression during cardiac reprogramming. However, it is important to note that the pattern of chamber-specific protein expression is highly correlated with chamber identity defined by action potential [10]. It may not be fair to directly compare the phenotypes of iCMs generated in vivo with the counter parts in vitro, given that the fibroblast population and reprogramming environment between in vitro and in vivo cardiac reprogramming are substantially different (e.g., adult cardiac fibroblasts vs. MEFs, and ischemic and inflammatory condition vs. a highly controlled cell culture). Since types of fibroblasts and exogenous condition are known to affect cardiac reprogramming efficiency in vitro, it is possible that those can affect chamber specification during cardiac reprogramming.

## Figures and Tables

**Figure 1 cells-10-01513-f001:**
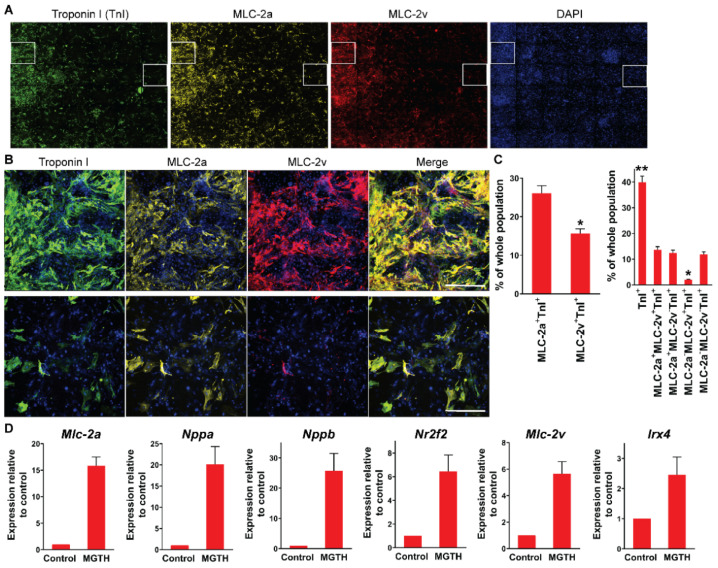
Incomplete chamber specification during in vitro cardiac reprogramming. (**A**) Composite immunofluorescent images used for high-content imaging analyses to quantify Troponin I, MLC-2a, and MLC-2v induction three weeks after MGTH transduction into MEFs. Each panel shows a composition of 36 images taken by the high-content imaging system using a 10x objective. (**B**) An enlarged single image of a white inlet in (**A**) (left inlet: top; right inlet: bottom). Scale bar, 400 µM. (**C**) Summary of high-content imaging analyses. *n* = 6. * *p* < 0.01 versus MLC-2a^+^TnI^+^ (left). * *p* < 0.01 or ** *p* < 0.001 versus MLC-2a^+^MLC-2v^+^TnI^+^ (right). MLC-2a^+^TnI^+^ = MLC-2a^+^MLC-2v^−^TnI^+^ + MLC-2a^+^MLC-2v^+^TnI^+^; MLC-2v^+^TnI^+^ = MLC-2a^−^MLC-2v^+^TnI^+^ + MLC-2a^+^MLC-2v^+^TnI^+^. (**D**) Expression profile of atrial and ventricular genes following in vitro cardiac reprogramming. Expression of indicated genes was quantified by qPCR three weeks after MGTH transduction into MEFs and normalized to uninfected control MEFs. Four independent experiments are presented as mean ± s.d.

**Figure 2 cells-10-01513-f002:**
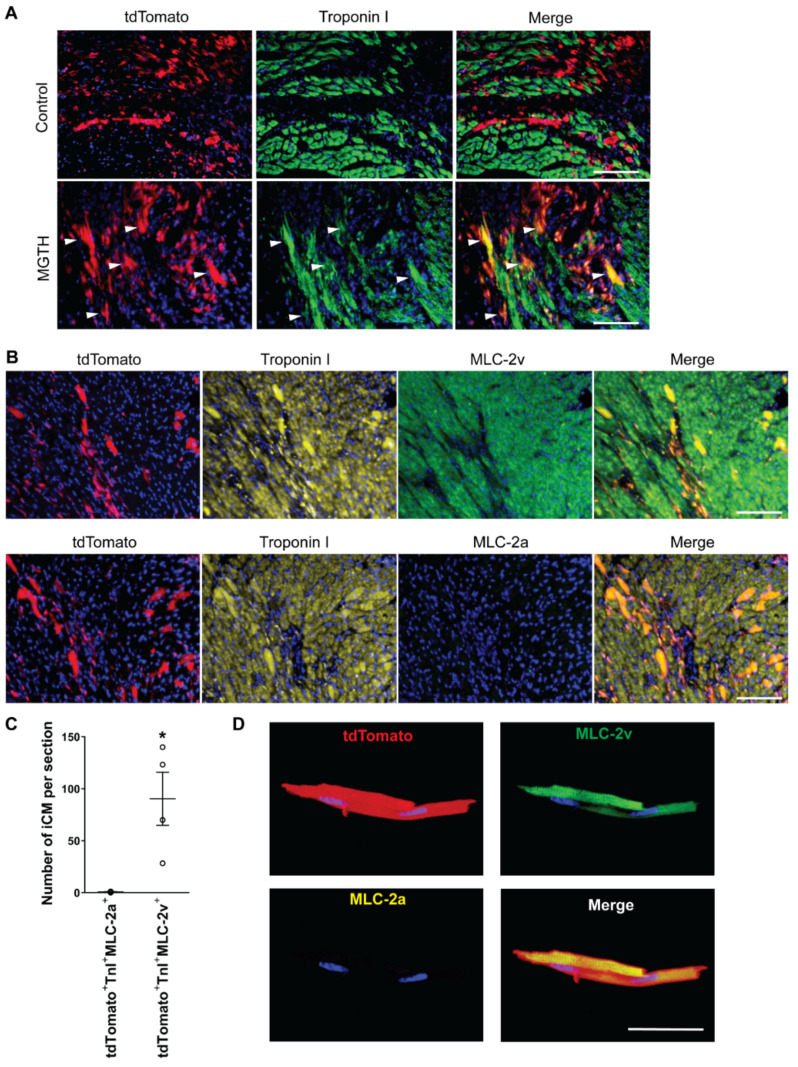
Exclusive induction of ventricular-like iCMs following in vivo cardiac reprogramming post-MI. (**A**) Lineage tracing of iCMs using Tcf21-iCre:R26R-tdTomato mice following MGTH or control vector injection post-MI. Heart sections were immunostained for Troponin I. White arrows indicate tdTomato^+^Troponin I^+^ iCMs. Scale bar, 100 µM. (**B**) Heart sections as described in (**A**) were immunostained for Troponin I and MLC-2v or MLC-2a. tdTomato^+^Troponin I^+^MLC-2v^+^ cells indicate ventricular-like iCMs. Scale bar, 100 µM. (**C**) Quantification of ventricular-like iCMs generated by in vivo cardiac reprogramming. The average number of iCMs from six heart sections per heart is presented. TnI: Troponin I. *n* = 4. * *p* < 0.05. (**D**) Isolated iCMs exhibiting a ventricular phenotype. Scale bar, 50 µM.

## Data Availability

Additional data are available from the corresponding author, Young-Jae Nam (young-jae.nam@vumc.org).

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
