# Peer review of "Chamber-Specific Protein Expression during Direct Cardiac Reprogramming"

_cells, 2021, doi:10.3390/cells10061513_

Round 1
Reviewer 1 Report
The manuscript by Zhang et al explores the chamber-specificity of cardiomyocytes generated by direct reprogramming of cardiac fibroblasts. The authors find that in vitro, the method produces mixed populations of atrial and ventricular cells with poor differentiated identities. However in vivo, ventricular cardiomyocytes are produced, which is desirable for therapeutic applications. Overall I think the manuscript is clearly presented and well organized and discussed, and I think it can be accepted after a minor revision (see below).
Major concerns
None
Minor Concerns
Figure 1C is very confusing to me. It seems like ~40% of cells reprogram successfully to iCMs (positive for TnI), which would leave 60% of cells as not reprogrammed fibroblasts. Then double positive cells are presented (TnI+/MLC2V+ and TNI+/MLC2a+), which makes sense, and then triple positive (TNI+/MLC2a+/MLC2v+), again makes sense and indicates undefined cardiomyocytes. The rest of the combinations presented do not make sense to me (TNI+/MLC2a+/MLC2v- is the same as TNI/MLC2a+ before, and so on). Furthermore the total sum of all categories is over 100% which does not make sense because we know only 40% are TNI positive. Overall, I think there is a misunderstanding in the way these data are presented and I hope the authors can rework this graph into one or more graphs that are more informative.
Author Response
We agree on the reviewer’s comment that the graph in Figure 1C is very confusing. As the reviewer pointed out, the sum of all data set is more than 100%, because MLC2a+TnI+ or MLC2v+TnI+ population includes MLC2a+MLC2v+TnI+ cell population.
TnI+ cells = (MLC2a+MLC2v-TnI+) + (MLC2a-MLC2v+TnI+) + (MLC2a+MLC2v+TnI+) + (MLC2a-MLC2v-TnI+)
MLC2a+TnI+ = (MLC2a+MLC2v-TnI+) + (MLC2a+MLC2v+TnI+)
MLC2v+TnI+ = (MLC2a-MLC2v-TnI+) + (MLC2a+MLC2v+TnI+)
To avoid the confusion, we provided two separate graphs in Figure 1C.
Reviewer 2 Report
The authors assessed the ability of the cardiac re-programming approach for chamber specification in vitro and in vivo. One of the study strengths is that this approach is in line with the efforts of the scientific community aiming to restore heart function after episodes of myocardial infarction and heart failure. Also, the authors demonstrated that in vivo cardiac reprogramming may have an inherent advantage of generating chamber matched new cardiomyocytes as a potential heart regenerative approach. However, the authors do not show any mechanism why this happen in vivo and not in vitro. In addition, although in vivo cardiac reprogramming may represent a potential heart regenerative approach, how can this be translated to humans? All these questions can be pointed out as weakness of the study.
Nevertheless, the design of the study agrees with the goals the authors propose for the study. All the experiments are well planned, and the methods are adequate for the study. qPCR experiments for atrial and ventricular markers could better support the protein data provided by the authors. Additionally, functional characterization of both types of cardiomyocytes would be interesting and could bring added value to the conclusions.
The English wording and grammar are in agreement with international standards.
Author Response
We performed qPCR experiments using atrial and ventricular genes. The results were provided in Figure 1D. We agree that functional characterization of iCMs would strengthen our conclusion. However, the main goal of this study is to investigate chamber specific protein expression during in vitro and in vivo cardiac reprogramming. In addition, we previously showed that subtype specific protein expression is highly correlated with its corresponding action potential pattern (Nam et al, Development 2014). We discussed this weakness in the last paragraph of Discussion.